# *HOXA* Amplification Defines a Genetically Distinct Subset of Angiosarcomas

**DOI:** 10.3390/biom12081124

**Published:** 2022-08-16

**Authors:** Hongbo M. Xie, Kathrin M. Bernt

**Affiliations:** 1Division of Pediatric Oncology, Department of Pediatrics, Center for Childhood Cancer Research, Children’s Hospital of Philadelphia, 3501 Civic Center Boulevard, CTRB 3064, Philadelphia, PA 19104, USA; 2Department of Bioinformatics and Health Informatics (DBHI), Children’s Hospital of Philadelphia, Philadelphia, PA 19104, USA; 3Department of Pediatrics, Perelman School of Medicine at the University of Pennsylvania, Philadelphia, PA 19104, USA; 4Abramson Cancer Center, Philadelphia, PA 19106, USA

**Keywords:** *HOXA*, homeobox, angiosarcoma, genomic

## Abstract

Angiosarcoma is a rare, devastating malignancy with few curative options for disseminated disease. We analyzed a recently published genomic data set of 48 angiosarcomas and noticed recurrent amplifications of *HOXA*-cluster genes in 33% of patients. *HOXA* genes are master regulators of embryonic vascular development and adult neovascularization, which provides a molecular rationale to suspect that amplified *HOXA* genes act as oncogenes in angiosarcoma. *HOXA* amplifications typically affected multiple pro-angiogenic *HOXA* genes and co-occurred with amplifications of *CD36* and *KDR,* whereas the overall mutation rate in these tumors was relatively low. *HOXA* amplifications were found most commonly in angiosarcomas located in the breast and were rare in angiosarcomas arising in sun-exposed areas on the head, neck, face and scalp. Our data suggest that *HOXA*-amplified angiosarcoma is a distinct molecular subgroup. Efforts to develop therapies targeting oncogenic *HOX* gene expression in AML and other sarcomas may have relevance for *HOXA*-amplified angiosarcoma.

## 1. Introduction

Angiosarcoma is a rare malignancy with a poor prognosis. The 5-year survival rates are about 30% with conventional approaches, and survival is almost entirely dependent on surgical resection of low-stage disease [1,2].

Due to the low incidence of angiosarcoma, large genomic studies have been challenging. One of the largest cohorts reported to date includes 48 samples from 36 patients who underwent comprehensive genomic profiling of archived tumor samples [3]. This study reported substantial heterogeneity in the mutational landscape of angiosarcoma related to location. Tumors from sun-exposed sites (head, neck, face and scalp (HNFS)) showed a pattern of high mutational burden that was associated with sensitivity to checkpoint blockade in two patients. The study also confirmed common alterations of the p53 and RAS pathways, consistent with previous reports [1,4,5,6], as well as a known risk of angiosarcoma in Li Fraumeni patients. The combination of p53 mutations and Ras pathway alterations in angiosarcoma was modeled in recent independent studies, and these tumors may be sensitive to mTOR or Ras pathway inhibition [6,7,8,9]. Recurrent mutations were also found in *PIK3CA,* MAPK signaling, *PTPRB, PLCG1, KDM6A* and amplification of *FLT*4 and *MYC* [4,5,10,11,12].

We reanalyzed the data from the study by Painter and colleagues [3] available in cBioPortal [13,14] in order to determine if additional subsets could be identified that might have potential therapeutic implications. We noticed an unexpectedly high frequency of *HOXA*-cluster alterations. The *HOX* genes are a set of 39 evolutionary conserved transcription factors that are arranged in 4 clusters (*HOXA, HOXB, HOXC* and *HOXD*) and determine body patterning during embryogenesis (Figure 1A) (reviewed in [15]). Paralogs within the different clusters exhibit a certain degree of functional redundancy, and in the right context, they can be swapped with only minor phenotypic consequences. *HOX* clusters are enriched for ultraconserved elements, which are frequently amplified in cancer and may contribute to oncogenesis through a variety of mechanisms [16,17]. The *HOXA* cluster in particular plays a role in the emergence of the blood-forming and vascular systems, which arise from a common embryonic progenitor [15,18]. Postnatally, the later *HOXA*-cluster genes (*HOXA7-13*) are expressed and required for the maintenance and growth of hematopoietic stem and uncommitted progenitor cells. Individual *HOXA* genes also play a role in regulating the emergence of endothelial progenitor cells (EPCs) from the bone marrow, neovascularization and wound healing [15]. This suggests a potential mechanism for *HOXA* amplifications to act as an oncogene in angiosarcoma. In this study, we report the molecular and clinical features of *HOXA*-amplified angiosarcomas in the patient cohort reported by Painter and colleagues [3].

### Methodology

Previously published and openly available data from the Angiosarcoma Project (Provisional, September 2018) were retrieved and analyzed based on Painter et al. 2020 [3]. *HOXA* cluster, *KDR* and *CD36* gene copy number (CN) alterations were inferred using ReCapSeg, as detailed by Painter et al. 2020 [3]. Clinical annotation data of angiosarcoma patients were also obtained from cBioPortal. Integrative genome viewer (IGV) tracks of the *HOXA* cluster in all 48 samples in Figure 1B,C show patients ordered by *HOXA3* sequence abundance. 

Significance testing for *HOXA* gene enrichment analyses in Table 1 used the Fisher’s exact test implemented in R (version 3.5). Significance was tested both on a sample level and a patient level, and results were similar for both. The more stringent patient-level analysis is shown. Bonferroni correction method as implemented in R (p.adjust() function) was applied to correct for multiple testing. Significance testing for CN fold-change (downloaded from cBioPortal) of *HOXA3* (as a surrogate for *HOXA* cluster amplification), *KDR* and *CD36* was performed using the Mann–Whitney U test.

BioVenn (biovenn.nl) was used to visualize the overlap of *HOXA*, *KDR* and *CD36* amplifications in angiosarcoma patients.

## 2. Results

Overall, amplification of at least one pro-angiogenic *HOXA*-cluster gene was present in 12/36 (33%) patients. The most frequently amplified *HOXA*-cluster gene with a documented role in vasculogenesis/angiogenesis was *HOXA3*. *HOXA9, HOXA11* and *HOXA13* were significantly co-amplified with *HOXA3* as part of a larger amplification (Figure 1B,C, left panel). In four patients, the amplification occurred within the *HOXA*-cluster encompassing a minimal region including *HOXA3-HOXA9*. 

*HOXA* amplification was significantly associated with amplifications of the vascular endothelial marker *CD36* (*p* = 5.59 × 10^−4^, Fisher’s exact test, remaining significant after Bonferroni correction at 1.62 × 10^−2^), and *KDR* (*p* = 1.07 × 10^−3^, Fisher’s exact test, remaining significant after Bonferroni correction at 3.11 × 10^−2^) (Table 1). The CN fold-change for both *KDR* and *CD36* was significantly higher in HOXA-amplified samples at *p* < 1 × 10^−^^5^ and *p* = 3.6 × 10^−^^4^, respectively (Figure 1C, Mann–Whitney U test). Similarly, the CN fold-change of *HOXA3* (as a marker for *HOXA*-cluster amplification) was significantly higher in *KDR* (*p* = 3.6 × 10^−^^3^) and *CD36* (*p* = 4 × 10^−^^3^) amplified samples (Mann–Whitney U test). There was substantial overlap between *HOXA*, *KDR* and *CD36* amplifications (Figure 1D). *KDR* encodes the VEGFR2, and *KDR* mutation/amplification has previously been identified by other groups [3,10,11]. Overall, *HOXA*-amplified tumors had a comparatively low mutation burden, although the difference failed to reach statistical significance (Figure 1E, Table 1). 

The percent of co-occurring signaling pathway mutations (*PIK3CA, MAP3K13*, *NF1, NRAS, HRAS*, *BRAF, RAF1*) was not substantially different from non-*HOXA*-amplified samples. RAS pathway mutations occur with similar frequencies in *HOXA*-cluster dysregulated AML, where they are usually subclonal. *TP53* alterations were also similar between the two groups (Table 1). 

We did not observe any statistically significant differences in the clinical presentation of *HOXA*-cluster-amplified angiosarcomas compared to those without *HOXA* amplification, although we noted several interesting trends that may warrant further study. We observed a trend toward a less frequent location in the sun-exposed HNFS area compared to the whole cohort (17 vs. 33% in those with and without *HOXA* amplification, respectively), and a more frequent location in the breast (67 vs. 46%). There was also a trend toward a lower percentage of male patients with *HOXA* amplifications (male: 17 vs. 33%). We observed a trend toward a higher proportion of patients receiving tumor-directed therapy at the time of sample submission (58 vs. 42%), which could be a surrogate for higher-stage disease linked to worse outcomes. However, reliable outcomes data for this cohort were not available. Based on the available clinical annotation in cBioPortal, only two patients with *HOXA*-amplified angiosarcoma had prior radiation. This seems low based on the clinical experience of breast angiosarcomas as secondary malignancies after irradiation for breast cancer, but is not significantly different from the entire cohort (17 vs. 21%). *HOXA*-amplified samples had the same age distribution, with two peaks: one around age 30, and a second peak around age 65 (Table 1).

**Figure 1 biomolecules-12-01124-f001:**
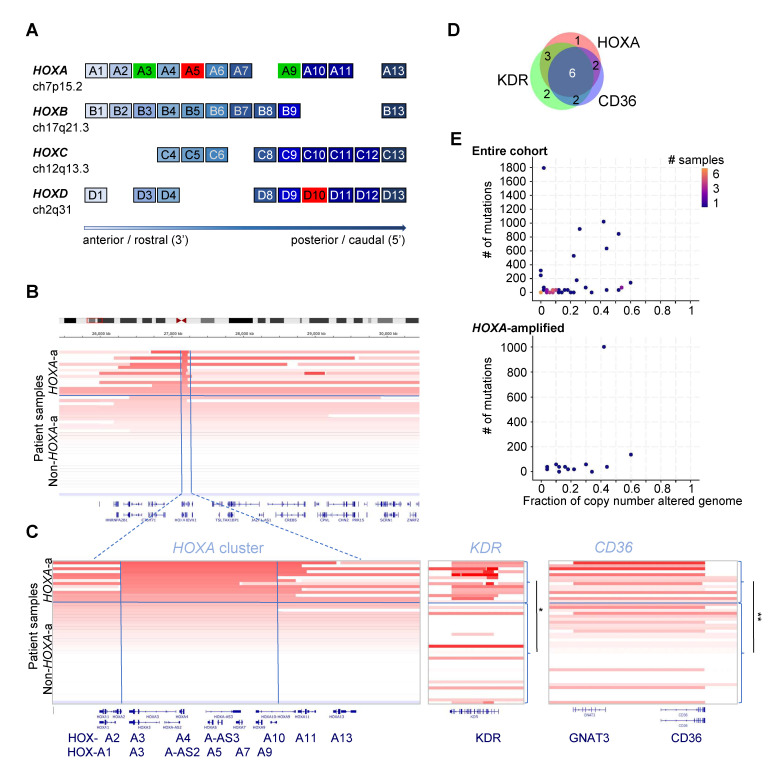
(**A**) Schematic of the human *HOX* clusters. Paralogs have the same number and coloring. The two key *HOX*-cluster genes that control EPC fate decisions and vasculogenesis, *HOXA3* and *HOXA9*, are marked in green. Anti-angiogenic *HOXA5* and *HOXD10* are marked in red. (**B**) *HOXA*-cluster amplifications in angiosarcoma. Out of 48 patient samples (12 out of 36 patients), 13 had amplifications in *HOXA3.* Most *HOXA3* amplifications also encompassed additional 5′ *HOXA* genes, including *HOXA9* (*HOXA3 to HOXA9* marked by vertical lines in (**B**), *p*-value: 1.9 × 10^−^^8^, p-adjust: 1.398 × 10^−^^7^), *HOXA* 11 (*p*-value: 2.596515 × 10^−^^7^, p-adjust: 1.817561 × 10^−^^6^) and *HOXA13* (*p*-value: 9.487666 × 10^−^^5^, p-adjust: 6.641366 × 10^−^^4^). Copy numbers (CNs) were inferred using ReCapSeg, as described by Painter et al. [3]. Statistical significance was calculated using Fisher’s exact test (Bonferroni correction). Visualization was performed using the IVG in cBioPortal sorting on fold-change of the *HOXA3* copy number. (**C**) *HOXA*-cluster, *KDR and CD36* amplifications in angiosarcoma patient samples. Visualization of CN fold-change of the *HOXA* cluster, *KDR* and *CD36* sorted on HOXA3-CN. * *p* < 1 × 10^−^^5^, ** *p* = 3.6 × 10^−^^4^, Mann–Whitney U test. (**D**) Venn diagram of *HOXA*-cluster, *KDR* and *CD36* amplifications in angiosarcoma patients. (**E**) # of mutations (y-axis) and copy number alterations (x-axis) of the entire cohort (top panel) compared to *HOXA*-amplified angiosarcoma (bottom panel) using the visualization tools in cBioPortal. The difference was not statistically significant.

**Table 1 biomolecules-12-01124-t001:** Patient clinical characteristics and recurrent amplifications/mutation of *HOXA*-amplified and non-*HOXA*-amplified angiosarcomas. *HOXA* amplifications were significantly associated with amplifications of *KDR* and *CD36*. There is also a non-statistically significant trend toward lower mutational burden and location in the breast versus HNSF. FE: fixed effect; MV: multi-variant.

Patient Characteristics (36 Patients Total)	*HOXA*-Amplified (12) # (%)	Non-*HOXA*-Amp (24) # (%)	Significance *p*-Value FE (MV)
**Gender**			
Male	2 (16.7)	8 (33.3)	ns
Female	10 (83.3)	16 (66.7)	ns
**Location**			
HNFS	2 (16.7)	8 (33.3)	ns
Breast	8 (66.7)	11 (45.8)	ns
intrathoracic/abd	2 (16.7)	4 (16.7)	ns
LIMB	0 (0)	1 (4.2)	ns
**Prior Radiation**			
Yes	2 (16.7)	5 (20.8)	ns
**Age**			
Peak 1: 26–40 Years	5 (41.6)	12 (50)	ns
41–50 Years	0 (0)	2 (8.3)	ns
Peak 2: 51–70 Years	6 (50)	8 (33.3)	ns
>70 Years	1 (8.3)	2 (8.3)	ns
**Pathology**			
Vasoformative	10 (83.3)	24 (100)	ns
Epithelioid	7 (58.3)	8 (33.3)	ns
Spindle Cell	6 (50)	17 (70.8)	ns
**Mutation Burden**			
>200 Mutations	1 (8.3)	7 (29.2)	ns
**Amplifications**			
**CD36**	**8 (66.7)**	**2 (8.3)**	**5.59 × 10^−^^4^ (1.62 × 10^−^^2^)**
**KDR**	**9 (75)**	**4 (16.7)**	**1.07 × 10^−^^3^ (3.11 × 10^−^^2^)**
PHF1	9(75)	10 (41.7)	ns
**Mutations**			
TP53	4 (33.3)	7 (29.2)	ns
KDR	3 (25)	5 (20.8)	ns
PIK3CA	2 (16.7)	4 (16.7)	ns
NRAS	0	3 (12.5)	ns
HRAS	1 (8.3)	1 (4.1)	ns
BRAF	0	2 (8.3)	ns
RAF1	0	1 (4.1)	ns
NF1	2 (16.7)	2 (8.3)	ns
MAP3K13	0	1 (4.1)	ns

## 3. Discussion

Here, we report a not previously recognized, high percentage of *HOXA* amplifications in angiosarcoma. Dysregulation of *HOXA*-cluster genes is a common feature of acute myeloid leukemia (AML). Distinct *HOXA*-cluster genes also play critical roles in regulating bone-marrow-derived endothelial precursor cells (EPCs) and in situ neovascularization [15]. Specifically, *HOXA3* and *HOXA9* are the two key angiogenic/vasculogenic homeobox transcription factors during development and in wound healing [19,20,21,22,23,24,25,26,27]. Furthermore, *HOXA11* is expressed in vascular smooth muscle cells in fetal but not adult tissue [28]. *HOXA* genes are separated into two distinct groups, the anterior (*HOXA 1–5*) and posterior (HOXA *7–13*) clusters. All *HOX* genes within each group are regulated coordinately in muscle development (controlled by the anterior *HOXA* cluster) and hematopoiesis (controlled by the posterior *HOXA* cluster). In contrast, individual *HOXA* genes within the anterior *HOXA* cluster have opposing functions in angiogenesis: *HOXA3* is a key pro-angiogenic transcription factor and is not expressed in adult quiescent vascular endothelium. Conversely, its close neighbor *HOXA5* is highly expressed in quiescent endothelium, and *HOXA5* expressing endothelial cells fail to respond to angiogenic stimuli [29,30]. In addition, *HOXD10* expression has been shown to inhibit angiogenesis [15,31].

*HOXA* genes play a critical role in the transformation of several subtypes or AML, suggesting that *HOXA* amplifications could also act as an oncogene in angiosarcoma. This is supported by the following observations: *HOXA* amplifications are common in angiosarcoma. Several of the amplified *HOXA* genes have well-described roles in driving blood vessel growth, suggesting a molecular mechanism for how *HOXA* amplifications could induce aberrant vascular proliferation. Finally, in a subset of patients, the amplified domain did not encompass the entire *HOXA* cluster, but there was consistent focal amplification of a segment that included pro-angiogenic members of the *HOXA* cluster (*HOXA 3–9*). 

Our analysis also suggests that *HOXA*-amplified angiosarcomas may have a different etiology from the more common angiosarcomas of the head, neck, face and scalp [2]. The latter carries a mutational signature that is consistent with UV-induced DNA damage [3]. In contrast, *HOXA*-amplified angiosarcomas may be caused by the amplification of developmental master regulators of vasculogenesis and angiogenesis. The observed trend of different presentations (deeper tumors, commonly in the breast versus sun-exposed areas) and a distinct genomic landscape (strong association with *KDR* and *CD36* amplification, as well as lower mutational burden) support the notion of a different origin.

Finally, our study raises the possibility that therapies developed to modulate *HOX*-cluster expression in other malignancies could have efficacy in this subtype of angiosarcoma. Regulators of *HOXA* transcription in other malignancies include Menin [32] and DOT1L [33,34,35]. Menin inhibitors in particular have reported pre-clinical efficacy in AML, Ewing sarcoma [36] and hepatocellular carcinomas [37], and are currently in clinical trials. Direct small-molecule inhibitors of HOX transcription factors are also an active area of research. 

Our study has several important limitations. Although this is the largest cohort of angiosarcoma patients reported to date, the number of patients is still quite small, and we were unable to conduct an independent validation. It will be critical to confirm *HOXA* amplifications in a second, independent, patient cohort. This may also allow confirming the amplification using a second independent method such as fluorescence in situ hybridization (FISH), and to better delineate differences in the mutational landscape, clinical characteristics, and outcomes of *HOXA*-amplified angiosarcoma. Functional mechanistic studies will be required to determine whether *HOXA* amplifications are in fact oncogenic, whether *HOX* gene expression is required for tumor growth, and whether it can be targeted for therapeutic purposes.

## Data Availability

All data are available at cBioPortal: http://www.cbioportal.org/study/summary?id=angs_project_painter_2018#summary [38].

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
