# Peer review of "HOXA* Amplification Defines a Genetically Distinct Subset of Angiosarcomas"

_biomolecules, 2022, doi:10.3390/biom12081124_

Round 1

Reviewer 1 Report

Overall a clearly written article that adds important information to the published results of the Count Me In project in angiosarcoma.

Points to be considered by the authors to improve the paper:

Table 1:

1.       Please expand FE and MV used in the table column heading

2.       Is there a particular rationale for using 10-year groupings for age? As age had 2 peaks in the cohort, it could be categorized into 3 groups taking into note the cutoff for the peaks.

Overall:

1.       There are a few minor errors in grammar and spelling that need to be corrected.

Author Response

Thank you for the positive review.

  1. Thank you for noticing that the abbreviations FE and MV need to be spelled out. We have included the full spelling for FE and MV in the legend of table 1.
  2. We agree that shortening the table and combining age groups in the two peak age brackets makes sense. We have done so in the revised manuscript.
  3. We have carefully checked spelling and grammar throughout the manuscript.

Reviewer 2 Report

This is an interesting re-analysis of a publicly available dataset of DNA sequencing for angiosarcoma. The authors describe the finding of HOXA amplification in a subset of tumors. The statement “We propose here that HOXA amplifications act as an oncogene in angiosarcoma” is not strongly supported solely by the finding of amplification in a small subset of these tumors and should be removed, in favor of a statement with more uncertainty given the relative paucity of data. This finding should be further investigated to determine its biological importance and clinical utility.

There are several typographical errors that need to be corrected:

-        Typo in introduction “Tumors from sun exposed sited”

-        In methodology, the S in Angiosarcoma should not be capitalized

-        Correct the spelling of spindle in Table 1

-        There are several typos in the second paragraph of the discussion

Author Response

We fully agree that our observation is too preliminary to conclude that HOXA amplifications are an oncogenic event in angiosarcoma. We have moderated the statement highlighted by the reviewer. We also agree that further studies – both a second independent cohort, as well as mechanistic studies, are needed. We have added a statement to this effect at the end of the discussion.

Thank you very much for finding some of the typographical errors – these were corrected

Reviewer 3 Report

Dear Editor, 

The manuscripts from Bernt and colleagues reported in silicon analysis of a high percentage of HOXA amplifications in a sub-class of angiosarcomas associated with amplification of KDR and CD36  suggesting a different etiology of this sub class.

Although, dysregulation of HOXA cluster genes is a common feature of acute myeloid leukemia (AML). Distinct HOXA cluster genes also play critical roles in regulating bone marrow-derived endothelial precursor cells (EPC) and in situ neovascularization . The authors reported for the first time this amplification in angiosarcomas.

However, 

1-the paper can be improved by completing  the genomic characterization with a graphical representation of KDR and CD36 fold change in this sub class

2-reporting the graphical distribution of HOXA genes in different groups and analyzing with Mann Whitney U-test  considering the significant differences 

3- analyzing for this amplification of new fresh tissues

Author Response

  1. The paper can be improved by completing  the genomic characterization with a graphical representation of KDR and CD36 fold change in this sub class. Response: We agree that a visual representation of the KDR and CD36 amplifications would be helpful. We have added it to Figure 1C and report the significance of the difference in CN fold change using the   Mann Whitney U-test (see question 2 below).

  1. [The paper can be improved by] reporting the graphical distribution of HOXA genes in different groups and analyzing with Mann Whitney U-test  considering the significant differences. Response: Given the substantial overlap of the three groups (HOXA, KDR and CD36 amplified), it seems a bit redundant to duplicate Figure 1C to show the same data two more times, once sorted on KDR and then sorted on CD36. However, we have interrogated the level of significance of the CN fold change of HOXA3 (as a surrogate to HOXA cluster amplification) in KDR amplified vs non amplified, and CD36 amplified vsw non amplified samples using the Mann Whitney U-test. We also added a Venn diagram that will show the extent to which all three alterations co-occurr (Figure 1D).

  1. [The paper can be improved by] analyzing for this amplification of new fresh tissues. Response: We agree that it will be critical to confirm these findings in an independent cohort. Unfortunately, we do not a large enough local cohort. We reached out to the authors of the second largest cohort of angiosarcomas that underwent a snp array published to date (Andreas Braeuninger - 10 samples, two with KDR amplification). Unfortunately, we did not receive a response, and the data is not publicly available.  In the revised manuscript we now include a paragraph explicitly discussing this important limitation.

Round 2

Reviewer 3 Report

Dear Editor ,

the authors addressed all my points

I asked to add this reference in the introduction doi: 10.1016/j.bbcan.2017.09.001.

and minor typing errors in the text

Author Response

Thank you for this suggestion. We included a description of ultraconserved elements, their enrichment in HOX clusters and their frequent amplification in cancer, and we cited the very nice review by Terracciano and colleagues.
